# The Antioxidant Activity of Mistletoes (*Viscum album* and Other Species)

**DOI:** 10.3390/plants12142707

**Published:** 2023-07-20

**Authors:** Marcello Nicoletti

**Affiliations:** Department of Environmental Biology, Foundation in Unam Sapientiam, Sapienza University of Rome, 00185 Rome, Italy; marcello.nicoletti@uniroma1.it

**Keywords:** mistletoe, *Viscum*, antioxidant activity, Santalaceae, flavonoids

## Abstract

In addition to the European mistletoe, *Viscum album*, which is the most known and utilized one, there are several species commonly known as mistletoe. They are spread in various regions of the planet and are all characterized by hemiparasitism and epiphytic behaviour. The published studies evidence other similarities, including the sharing of important biological properties, with the common presence of antioxidant effects. However, whereas the European mistletoe is largely utilized in medical treatments, although with controversial aspects, the scientific knowledge and medical uses of other mistletoes are still insufficient. This review focuses on the controversial medical story of European mistletoe regarding its antioxidant activity and the potentiality of the other species named mistletoe pertaining to botanical families and genera different from *Viscum*.

## 1. Introduction

This review focuses on research concerning the antioxidant properties of mistletoes and considers their utilization in several fields and related pathologies. Mistletoes are special plants: unique in behaviour, chemical content, and metabolic processes, with a controversial human interest and utilization [1]. Their recent use in medicine as a complementary cancer treatment [2,3] is mainly based on evidence of general effects, like the antioxidant [4], anti-inflammatory [5], and immune-adjuvant [6] properties. Therefore, the literature on mistletoe extracts is an expanding area of research and now extends to the study of non-European species. Current studies vary from the safety of the extracts [7] to their utilization in several sectors of herbal interest, however, the main focus concerns metabolic stress.

The impact of chronic and long-term degenerative diseases on society will increase due to the elongation of life expectancy. Inflammation, oxidative stress, and immune responses are deeply involved in the current major causes of death, such as cardiovascular diseases and cancers [8,9,10]. Among the proposed solutions to protect against oxidative damage from biomolecules is the utilization of natural substances [11].

The reasons to consider natural products as potential treatments for chronic diseases were reported in the study by Dančík et al., based on computational evidence [12]. In the conclusion, they state: “our results indicate that targets of natural products are highly connected, much more so than genes implicated in human disease, … natural products tend to target proteins more essential to an organism than are disease genes”. In other words, natural products target proteins, which are more closely connected to the cellular and organism network than the gene targets implicated in human diseases, and therefore natural products affect more general protein networks essential to metabolism.

The reports on the antioxidant activity of mistletoe species are reported and discussed here, taking into consideration their multiple effects and controversial history. The traditional use of herbs implies that most of the attention will be directed towards the species of the Old Continent, as is the case with the *Viscum album*. However, there are other similar mistletoes worldwide, which could have similar or improved effects, as evidenced by the research. The controversial parabola of European mistletoe, from a poisonous plant to an ingredient of many drugs, is still ongoing. Its main tendencies, including utilization against chronic diseases in the elderly, are evidenced here as a platform for further steps. This review completes the previous one published in this journal concerning the anti-inflammatory activity of the *Viscum album* [13].

The references present in this review are the result of an extensive literature survey, comprehensive of more than 400 publications. Reports were obtained mainly on Google, PubMed, NCBI, and Scopus, following the Preferred Reporting Items for biological activity. The main keywords were “Viscum, mistletoe, antioxidant, anti-ROS, clinical trials, Santalaceae and related genera”.

### 1.1. The Mistletoe Plants

Plants commonly known as mistletoe correspond to different perennial dicotyledonous angiosperms species pertaining to the Rosales order. The most recognized mistletoe species are included in the genera *Viscum*, *Phoradendron*, *Arceuthobium*, *Peraxilla*, *Loranthus*, *Amylotheca*, *Amyema*, *Taxillus*, *Psittacanthus*, and *Scurrula*, which are spread throughout different botanical families. All of these are epiphytic species, meaning that they have no contact with soil, owing to their characteristic of hemiparasitism, which allows the mistletoes to extract raw sap from a host plant [14,15,16]. Parasitism of various types is present in at least 1% of plants but via different connections with the host [17]. Botanical studies, including a comparison between the physiology of mistletoe and the host, have expressed that, in the majority of cases, this is a symbiotic relationship in which there is an exchange of nutrients and a positive consequence for both species, except in conditions of drought, when there is strong competition for water [18,19,20]. As a consequence of this, mistletoe plants present a great variety of forms, shapes, and content, even between members of the same species, in accordance with habitat conditions and hosts, which influence the metabolome [21]. In 2023, a survey of the published research on the biological properties (hypotensive, anticancer, antimicrobial, analgesic, and anti-inflammatory capabilities) of the hydroalcoholic extracts of some species of *Viscum* have been reported [22], whereas, in this review, the antioxidant activity is evidenced.

### 1.2. The Viscum Mistletoes

The predominant variety of mistletoe plants is those belonging to the genus *Viscum*, which includes about 100 species (from 70 to 150, depending on the source of information) worldwide [23,24]. It is currently a member of the Santalaceae family, but it was previously considered a part of the Viscaceae and was originally part of the Loranthaceae. According to World Flora Online, there are 32 *Viscum* species across Africa, with around 30 of these exclusive to Madagascar [25]; they parasitize a variety of trees and shrubs of different families [26]. Some *Viscum* species are specialized to selected hosts, whereas others can be found in many different plants, such as *V. album* [27].

## 2. The European Mistletoe

*Viscum album* L., commonly known as the European mistletoe, is the most well-known and studied mistletoe species (Figure 1). It is present across the globe, mainly in the Northern Hemisphere, but very rarely in China and Australia [28]. It is historically related to Europe, where the very similar oak mistletoe, *Loranthus europaeus* Jacq., is also commonly seen. *Loranthus europaeus* can be differentiated from Viscum album as it is characterized by yellow berries, rather than white (Figure 2) [29,30]. This species should not be confused with *V. album*, as it has a very different chemical composition, as is well demonstrated by HPTLC analysis [31].

Due to specific and unique characteristics, the European mistletoe has always attracted the attention of humans, who have tended to attribute magical, beneficial, auspicious, and healing powers to this plant.

The berries are certainly the most distinctive and representative part of the European mistletoe because they are white (as evidenced by the epithet “*album*”) and very ostentatious, unlike the flowers, as evidenced in Figure 3. They are considered the most dangerous part of the whole plant, especially if ingested by mistake; however, this event is quite rare (as judged by related reports) [32]. Originating in the late Middle Ages, a period of obscurantism and prevention began, which had its main fulcrum in the toxicity of berries. Therefore, the use of mistletoe extract has fluctuated between ostracism from its use as an official medicine and the continued use in traditional medicines in all countries where the plant is present [33,34].

After a long period of discouragement of the use of mistletoe to avoid its toxic effects, a change occurred, characterized by the use of hydroalcoholic extracts in anthroposophic medicine to treat cancer [35,36]. This utilization continued uninterrupted and increasingly, mainly in Germany and Switzerland. Although many studies have clarified the mechanism of action, which allows mistletoe to cause selective cell cycle arrest in tumor cells, there are still many doubts about the ability of mistletoe extracts to fight cancer, as questioned by many authors. Meanwhile, the attention of clinicians has shifted to the collateral and complementary effects, both in support of the drug and against the impact of conventional treatments. The clinical trials reported immunoadjuvant, antioxidant, and anti-inflammatory properties [37], including the utilization of the extracts to improve the quality of life of the treated subjects [38]. In this scenario, the antioxidant activity plays a central role, together with the anti-inflammatory properties [39,40,41,42]. The most reported active constituents are known flavonoids, which are present in alcoholic and aqueous extracts [43,44] and are already reported for their anti-ROS and radical-scavengers properties [45], with attention to the variation of the phenolic content in relation to the host [46,47] and the investigated subspecies [48,49,50,51,52,53,54]. Several studies amplified the research to other constituents to evidence how they may cooperate with the activity. Among the constituents of *V. album*, in addition to microproteins, like viscoproteins and viscolectines [55,56], other constituents have been considered responsible for the activity [57,58,59,60,61,62].

Following the phytocomplex approach, in several papers, more research is considered [63] on the connection made between oxidative stress, inflammation, and many related pathologies [64], in particular, the hypoglycemic effects [65,66]. The antioxidant activity of *Viscum* extracts has been associated with hypoglycaemia, considering the important role played by oxidative stress in tissue damage by hypoglycaemia and diabetes [67,68,69,70,71], which may be the result of the deterioration in glucose homeostasis caused by metabolic disorders.

This is also in accordance with the possible utilization of *Viscum* sp. in anti-obesity products and other uses as a potential food and dietary supplement in the nutraceuticals market, which can be associated with the properties of ANFs, the ANtinutritional Factors. ANFs are substances of plant origin, which are considered to be able to reduce the absorption of nutrients, with effects on gastrointestinal function and some metabolic processes [72]. However, there are concerns about the utilization of ANFs, including lectins, saponins, and natural antioxidants, like phytic acid [73]. The negative effects could be reduced or eliminated by preliminary treatments, like fermentation and soaking, as already adopted in the utilization of several foods. It is noteworthy that most of the hydroalcoholic extracts currently used are already subjected to fermentation procedures, although they are administered by intradermal injection. Despite these concerns, the EMA (former EMEA) [74], as well as the US National Cancer Institute, have reported an absence of toxic or severe effects for the oral use of *Viscum* extracts [75].

## 3. The Asian Mistletoes

Korean mistletoe is the subspecies *coloratum* of *V. album.* It is commonly known as Korean mistletoe because of its wide distribution in many Asian countries, where almost all of its constituent parts are used in folk medicines. In particular, it is present in Ayurvedic and Chinese traditional medicines to treat many diseases, including inflammation, bone fractures, joint pains, arthritis, and hypertension [76]. It is adopted in the Pharmacopoeia of the People’s Republic of China. It can also be called *V. articulatum* Burn. f., and presents under many synonymous names, such as *V. aphyllum* Griff., *V. attenuatum* DC, *V, compressum* Poir, and others; it has also been identified as *Aspidixia articulata* (Burm. fil.) van Tiegh [13,77]. This is only one of the numerous related subspecies and varieties of *V. album*, spread across the planet and present in several tropical hosts, like the cultivated plants of cocoa, coffee, guava, rubber trees, and many others. Research into Asian mistletoes is limited in comparison with European mistletoe [21] but Korean mistletoe is quite an exception since it has been the object of several papers, reporting positive effects in cases of hypertension, ulcers, epilepsy, inflammation, wounds, and nephrotic disease [78]. Major evidence strongly suggests immunostimulant, anti-inflammatory, and antibacterial activity, confirming the properties already reported for the European mistletoe [79,80]. Among the confirmed activities was its antioxidant ability [81], which was associated with an incremental increase in the contents of caffeic acid and lyoniresinol [82]. The future utilization of *Viscum* extracts in nutraceuticals is also supported by recent research on the positive effects of Korean mistletoe on muscle performance, also in the elderly, including randomized trials [83,84,85,86,87,88]. The attention of Eastern researchers to the utilization of Korean mistletoe for food supplements cannot be considered insignificant since the functional foods originate in Asian countries, named in Japan as Foods for Specified Health Use (FOSHU), and now are present in many countries.

A potent antioxidant property has been reported for different extracts and bioactive constituents of this species. The antioxidant activity of the extracts was attributed to the presence of polyphenols, such as flavonoids and other phenols [89,90], also in consideration of their documented antioxidant activities [91]. Radical scavenging in vitro and in vivo properties of the ethanolic extract of the whole plant was attributed to the phenolic and triterpenes constituents, i.e., oleanolic acid and lupeol, as reported in Table 1. Several studies testify to the absence of toxicity and serious adverse effects. Among the constituents indicated as being responsible for the numerous activities, in addition to the known compounds, including oleanoic acid, betulinic acid, naringenin, and β-amyrin acetate, the compounds named visartisides merit special attention (Figure 4) [90,92,93].

Visartisides A-F are molecules with interesting structures based on a disaccharide nucleus linked to two aromatic units, one of phenylpropanoid type and one C6-C1, joined with an ester and an ether link, respectively. Some visartisides showed relevant anti-inflammatory and antioxidant activity [94,95]. They can inhibit NO production by macrophages, with an effect superior to that recorded for quercetin, which was used as a positive control. The same activity was shown for the ethanol extract of the whole plant when used against human erythrocytes, through stabilization of the cell membrane, with a maximum protection of 68% against haemolysis in comparison with a value of 72% for indomethacin, using a control standard. Although the plant contains flavonoids and polyphenols in good quantity, it is noteworthy that visartisides do not have phenolic groups or triterpene structure. Visartisides were also identified in the metabolome of Romanian *V. album* in a total of 140 metabolites pertaining to 15 secondary metabolite categories [96]. The identification of antioxidant agents present in plants affords an effective strategy to inhibit pathogenic processes resulting from exposure to numerous degradative syndromes, which are related to oxidative stress and DNA damage by ROS and RNS. Oxidative damage plays a significant role in many human pathologies, including mutagenesis, cancer, and aging, among others [97,98]. Therefore, constituents such as visartisides offer the possibility of obtaining a novel class of antioxidant compounds, although further studies concerning the mechanisms of action are necessary. In addition to *V. articulatum*, *V. liquidambaricolum* Hayata, another native Chinese mistletoe that is parasitic on *Pyrus*, was investigated for its antioxidant activity. Previous phytochemical investigations reported the presence of phenolic glycosides, flavanone glycosides, triterpenoids, organic acids, and flavonoids as the major secondary metabolites [99].

Among the Asian mistletoes, *Viscum monoicum* Roxb. ex DC. (syn. *V. heyneamum* DC.) merits a special mention. This species is present in several Asiatic countries, i.e., China and India, where it is utilized in traditional medicines to treat many ailments, i.e., pains, neuropharmacological disorders, and cancers. The analyses revealed relevant antioxidant and anti-inflammatory activities, which were assigned to polyphenolics and flavonoid constituents [100]. Another Indian mistletoe corresponds to *Dendrophthoe falcata* (L.f.) Ettingsh (syn. *Loranthus longifluorous*) (Loranthaceae) and is one of the most parasitic widespread plants in the deciduous forests of Western India. It contains flavonoids, triterpenes, and oleic acid. This species, together with *D. pentandra* (L.) Miq., has been used in the ethnomedicinal systems of India and Indonesia for the treatment of various ailments, including ulcers, asthma, paralysis, skin diseases, tuberculosis, and menstrual pain. *D. falcata* has been studied for its antimicrobial activity [101,102]. A review of 2023 reports shows studies on several properties, including the antioxidant effects through the inhibition of lipid peroxidation [103].

## 4. The American Mistletoes

In the southern states of the USA and Mexico, American mistletoe is found, also known as eastern mistletoe and hairy mistletoe. It is similar to the European mistletoe regarding the shape and fruits and corresponds to the species *Phoradendron leucarpum* (Raf.) Reveal & M.C. Jonst (Santalaceae, previously reported as *P. flavescens* and *P. serotinum*, and also transcripted as *P. leucoparcum*), and is present also as the subspecies *tomentosum*, named the hairy mistletoe. As European mistletoe, this species was considered toxic in the past, as it is associated with the presence of phoratoxin, which causes vasoconstriction. The toxicity may be the basis of its purported uses, however, it is also a concern for the medicinal use of *Viscum* sp. The concern was that high doses could induce delirium, hallucinations, bradycardia, hypertension, and cardiac arrest [104]. However, few reports of serious complications from accidental ingestion can be found, and adequate studies have instead demonstrated the contrary for the oral extract in several forms, which are purported to be beneficial for hypotension and constipation. Currently, in the scientific literature, after the pharmacological check, this species is considered free of serious negative or toxic effects [105,106]. This declaration of the safety of the American mistletoe is desirable to promote further study into the biological properties of this important species.

Among the Chilean mistletoes, *Tristerix tetrandus* (R. et Pav.) Barlow et Wiens (Loranthaceae) is traditionally used as an anxiolytic, anti-inflammatory, digestive, haemostatic, and hypocholesterolemic remedy. Its relevant antioxidant activity was reported and validated by the DPPH antiradical activity assay, the superoxide anion inhibition assay, and the ferric reducing activity, measured as micromoles of Trolox, as standard. The metabolome analysis was performed by UHPLC/MS-MS, evidencing the relevant presence of glycosidic anthocyanins and other flavonoids [107]. The Argentinian mistletoe, *Ligaria cuneifolia* (R. et P.) Tiegh. (Loranthaceae), is found in Chile, Peru and Argentina and is used in folk medicines. This plant is quite different from other mistletoes because of the evident, red, six, long-petaled with yellow-coloured fruits. Several studies have been performed on this species, reporting anti-hypertensive and anti-cholesterolemic actions, together with antioxidant and immunomodulatory properties [108]. The antioxidant activity of this plant was investigated in connection with the flavonoid-rich fraction [109]. The ex vivo and in vivo tests evidenced an encouraging inhibition of phospholipid oxidation and radical damage. Similar data are reported for *Phoradendron liga* (Gillies ex Hook. & Arn.) Eichler (Santalaceae), another Argentinian mistletoe which grows mainly in Bolivia and Brazil, in addition to North Argentina, preferably in dry tropical regions. It is noteworthy that triterpene betulin was detected as the active molecule responsible for the inhibition of the P-glycoprotein function in leukaemia cells, evidencing how it is necessary to investigate the active constituents of mistletoe plants [110,111].

## 5. The African Mistletoes

Mistletoes, in particular those from the genus *Viscum*, are largely present in Africa. However, these plants were poorly studied, albeit there are very interesting examples of variation and adaptation, including species parasitizing succulent plants, mainly present in Southern regions. The *V. minimum* Harv. is a very small mistletoe that lives inside the stem of the columnar great Euphorbiaceae. *V. minimum* has no leaves, roots or stems and lives inside the succulent cactus body of the host [112]. In this way, the plant can survive drought and remain protected. It is visible only during reproduction when the small flowers and fruits emerge. Another example is *V. crassula* Eckl. & Zeyh, which is a succulent mistletoe specialized in parasitizing *Portulaca afra*, another succulent species (Didieraceae), which is one of the favourite foods of elephants.

*V. rotundifolium* L. f. is a red-berry mistletoe, mainly present in southern Africa and able to survive in the Kalahari Desert [113]. The antioxidant and antibacterial activity of this hemiparasite was investigated together with its host, *Mystroxylon aethiopicum* (Thumb.) Loes subsp. *schlechteri* (Loes.) R.H. Archer (Celastraceae), commonly known as bushveld kooboob berry. The simplified phytochemical tests evidenced a similar chemical composition for the most common secondary metabolites, as well as the absence of relevant differences in the extracts with different solvents from the two species. Finally, it was possible to determine the absence of toxicity to human cells and the safety of the plant extracts [114,115].

The antioxidant activity of the investigated mistletoe species is reported in Table 1. The reported data in Table 1 merits some additional consideration concerning the methods. There is a large range of available tests for examining antioxidant activity, i.e., ORAC, HORAC, TRAP, TOSC, CUPRAC, FRAP, ABTs, and DPPH tests [116]. Several papers on the antioxidant and enzyme activity effects of *Viscum* extracts are based on a variety of tests. The DPPH assay is the most used due to its ease, but, in many cases, further testing is required to obtain more information about the type of activity and the mechanism of action. If a plant extract is subjected to ordinary antioxidant tests, the novelty should be the absence of activity. Plant metabolism usually involves an abundance of phenols, whose radical scavenger and anti-ROS properties are well known. However, phenol functions can be found in several classes of natural products with very different structures, from monocycle hydroquinones to polymeric tannins; other molecules cannot be excluded. Similar considerations can be presented for chemical analyses, which are used to provide evidence for the chemistry of the active constituents. The colorimetric Folin-Ciocalteu reagent is often employed to measure the number of phenolic compounds. However, this method could lead to erroneous results due to an overestimation of the true phenolic level in the analysed reagent, since the reagent reacts with all the oxidants present. Analogous consideration for the Dragendorff reagent, commonly used for the detection of alkaloids, which can react in the presence of compounds with an alpha, beta-unsaturated ketone group, such as in several phytosteroids.

Therefore, considering these factors, the argument must focus on the effectiveness and the mechanism of action. In the literature regarding the antioxidant activity of the European mistletoe, there are publications with such simplified tests, mainly focused on ethnobotanical studies [117,118,119], but most of the contributions are interested in connecting the activity with the mechanism or with other properties, including the study of the constituents responsible of the biological properties.

## 6. Conclusions

Table 1 contains the species discussed in this review, including the common names, the antioxidant studies, and the related references. It is evident that so far, the studies were influenced by the prevalent presence of European mistletoe and its utilization in complementary treatments in oncology [120,121,122], though there is an abundance of research focused on antioxidant activities. Southern African countries are a potential source of native, undiscovered, exceptional species [123]. The results of the metabolomic analyses, based on new, advanced analytical methods, are changing the previous information based on classic phytochemical methods [97,105,124] and evidence of the variation of the metabolome concerning the host and the harvest period [125]. In addition, the study of the interactions with protein networks is in progress due to in silico analyses [126]. In particular, the proteomic study provides evidence of many special microProteins, in addition to viscolectins and viscotoxins, with antioxidant activity [54,92,127,128,129,130,131]. Finally, this paper suggests that the chemical and pharmacologic value of mistletoes is only just being realised, and further research may reveal more interesting properties and other species to be investigated [132].

Concerning the future of mistletoe in medicine, the utilization of mistletoe extracts is increasing, in particular in oncology treatment, albeit always under discussion [133] and in consideration of the complementary effects, such as the antioxidant and immunostimulation properties [134]. However, the use of these extracts is also related to many other medical products, which are increasing their relevance in the market, like the utilization in homeopathy [135], nutraceuticals [136] meristemotherapy [31,137], and veterinary treatment [138,139].

## Figures and Tables

**Figure 1 plants-12-02707-f001:**
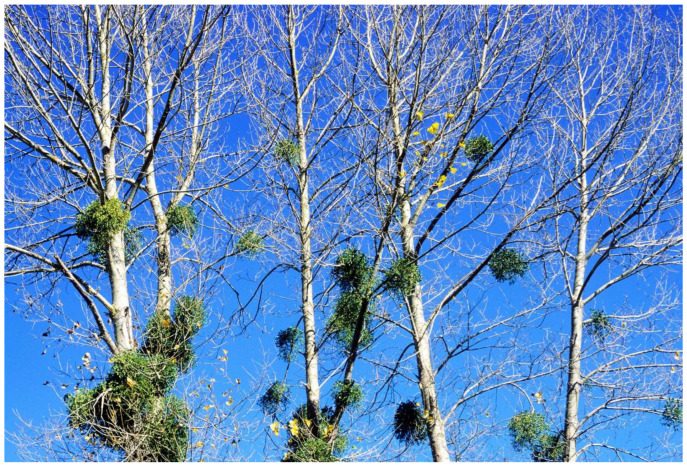
The European mistletoe is most visible in winter on a deciduous tree.

**Figure 2 plants-12-02707-f002:**
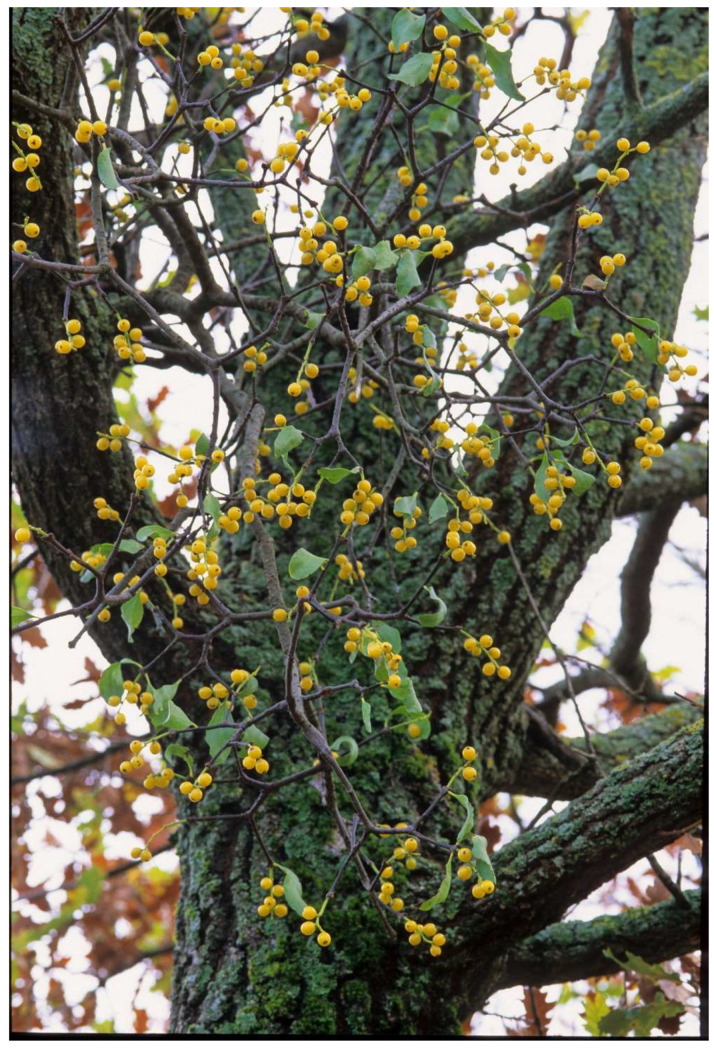
The oak mistletoe, *Loranthus europaeus*, with yellow berries.

**Figure 3 plants-12-02707-f003:**
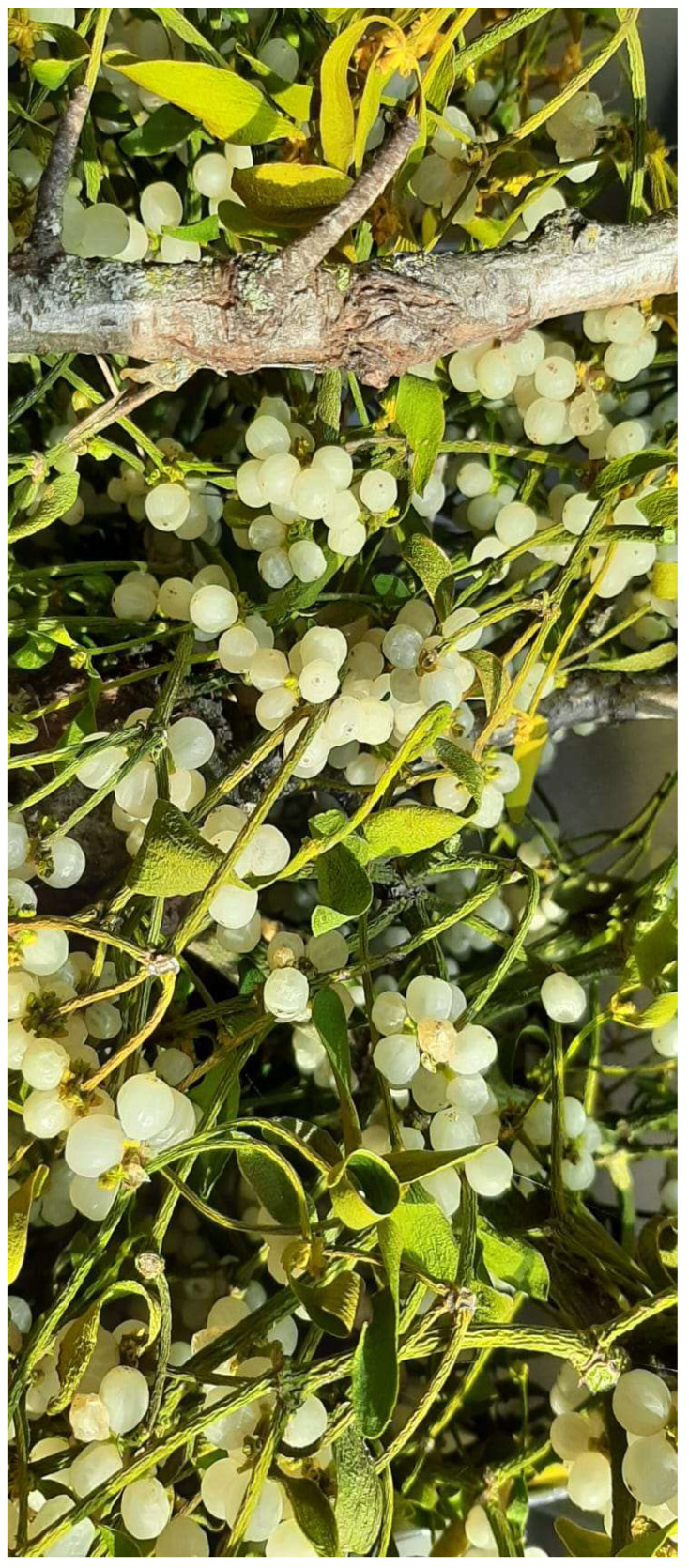
The berries of the European mistletoe are evident and abundant.

**Figure 4 plants-12-02707-f004:**
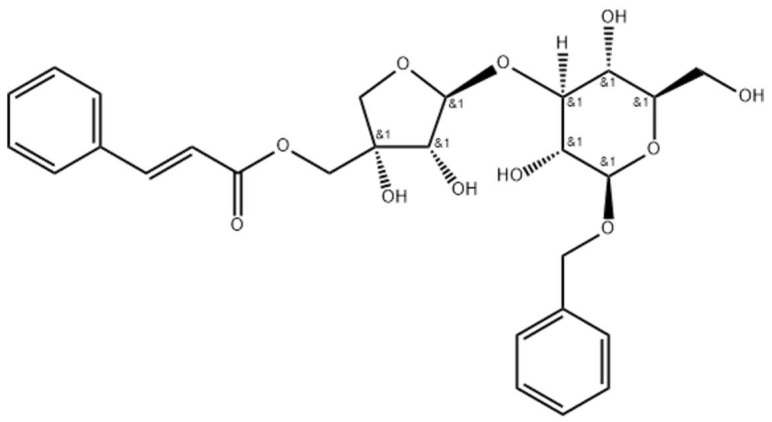
The structure of visartiside E, with evidence of the prochiral centers (&1) of the molecule.

**Table 1 plants-12-02707-t001:** Mistletoe species were investigated for their antioxidant activity, reported are the utilized tests and active constituents.

Mistletoe Species	Antioxidant Tests	Reported Active Constituents
European Mistletoe *Viscum album* L.	DPPH	Flavonoids
*Viscum album* L.	DPPH, FRAP	Several
*Viscum album* L.	Effect against oxidative damage	Polyphenols
*Viscum album* L.	Variation of enzymes activity (SOD, CAT, GTS)	Phenols
*Viscum album* L.	DPPH	Polyphenols and phenols
*Viscum album* L.	DPPH and enzymes activity (superoxide anion, HORAC, nitric oxide)	Polyphenols and phenols
*Viscum album* L.	enzymes activity (increase in proline content and in superoxide dismutase, ascorbate peroxidase, glutathione peroxidase, and glutathione reductase activity)	N.D.
*Viscum album* L.	DPPH, FRAP	Polyphenols and phenols
*Viscum album* L.	DPPH and enzymes activity (lipid peroxidation by ferric thiocyanate and hiobarbituric methods)	N.D.
*Viscum album* L.	enzymes activity (selectively inhibiting cytokine-induced expression of cyclooxygenase-2)	N.D.
*Viscum album* L.	DPPH, ABTS, FRAP	Quercetin, rosmarinic acid, catechin
*Viscum album* L.	DPPH, ABTS, AChE activity. total antioxidant capacity, lipid peroxidation assay, hydroxyl radical scavenging activity and enzymes activity (SOD, CAT, GSH-Px, GR, GST, lipid peroxidation levels)	Phenols and other
*Viscum album* L. subsp. *abietis*	DPPH	Phenols
Korean mistletoe *Viscum album* subsp. *coloratum* L.	DPPH	Caffeic acid, oleanoic acid, naringenin, esperetin, betulinic acid, syringaldehyde, and lyoniresinol
*Viscum album* subsp. *coloratum* L.	DPPH, superoxide radical scavenging	Flavonoids
*Viscum album* subsp. *coloratum* L.	radical scavenging activities toward DPPH and AzBTS-(NH_4_)***_2_***	MicroProtein
*Viscum album* subsp. *coloratum* L.	DPPH, FRAP, TPC	Phenols
Asian mistletoe *Viscum monoicum* Roxb. ex DC.	DPPH, ABTS	Polyphenols, flavonoids
*Viscum articulatum* Burn.	DPP and enzyme activity (lipopolysaccharide-induced nitric oxide assay)	Flavones, visartisides
*Viscum articulatum* Burn.	FRAP, HPSA, DPPH, DRSA and TEAC	Flavonoids
*Viscum liquidambaricolum* Hayata	FRAP, HPSA, DPPH, DRSA and TEAC	Flavonoids
*Viscum articulatum* Burn.	enzymes activity (inhibition effect on superoxide anion)	Flavanones
Indian mistletoe *Dendrophthoe falcata* (L.f.) Ettingsh	enzymes activity (inhibiting lipid peroxidation, reduced glutathione, superoxide dismutase levels and increasing CAT activity)	N.D.
Chilean mistletoe *Tristerix tetrandus* Mart.	DPPH, FRAP, SA	Phenols, flavonoids, anthocyanins
Argentinian mistletoe *Ligaria cuneifolia* (R. et P.) Tiegh.	DPPH	Flavonoids
African mistletoe *Viscum rotundifolium* L.f.	DPPH, FRAP	Phenols

## Data Availability

No new data were created.

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
