# Peer review of "The Antioxidant Activity of Mistletoes (Viscum album and Other Species)"

_plants, 2023, doi:10.3390/plants12142707_

Round 1
Reviewer 1 Report
The review of “The Antioxidant activity of mistletoes (Viscum album and related species)” for Plants MDPI. The topic of manuscript fits within the scope of the journal Plants MDPI.
Nevertheless, several problems/doubts should be solved before the manuscript is suitable to be published.
Major comments:
The major comment is related to the "Introduction". In general, the Introduction in this manuscript is somewhat weak. In the introduction after line 52 there is not enough information about relevance. I believe that it is necessary to write a relevance on the issue of studying mistletoes. This will help improve the quality of the manuscript.
Specific comments:
L. 21-24. This sentence should be supplemented with references to the literature.
L. 29-31. Please check throughout the manuscript for correct citations. The instructions for authors (https://www.mdpi.com/journal/plants/instructions) state the following: "In the text, reference numbers should be placed in square brackets [ ], and placed before the punctuation; for example [1], [1-3] or [1,3]."
L. 31-42. This sentence should be supplemented with references to the literature.
L. 49-52. Please edit the text typeface.
L. 92. and L. 112. Two sections of the manuscript use the title "The European mistletoe".
L. 136-137. This sentence should be supplemented with references to the literature.
L. 167-177. This sentence should be supplemented with references to the literature.
L. 215-216. Please give the meaning of the abbreviation "EMA".
L. 256. Please replace Figure 4 with a higher quality, higher resolution one.
L. 354. Please replace Figure 5 with a higher quality, higher resolution one.
L. 372. Please replace Figure 6 with a higher quality, higher resolution one.
Author Response
Major comments:
The major comment is related to the "Introduction". In general, the Introduction in this manuscript is somewhat weak. In the introduction after line 52 there is not enough information about relevance. I believe that it is necessary to write a relevance on the issue of studying mistletoes. This will help improve the quality of the manuscript.
Thanks for the suggestion. Several considerations have been added
Specific comments:
- 21-24. This sentence should be supplemented with references to the literature.
Added references
- 29-31. Please check throughout the manuscript for correct citations. The instructions for authors (https://www.mdpi.com/journal/plants/instructions) state the following: "In the text, reference numbers should be placed in square brackets [ ], and placed before the punctuation; for example [1], [1-3] or [1,3]."
Changed
- 31-42. This sentence should be supplemented with references to the literature.
It is supported by the reference [3]
- 49-52. Please edit the text typeface.
Changed
- 92. and L. 112. Two sections of the manuscript use the title "The European mistletoe".
Changed
- 136-137. This sentence should be supplemented with references to the literature.
Added
- 167-177. This sentence should be supplemented with references to the literature.
Added
- 215-216. Please give the meaning of the abbreviation "EMA".
A list of abbreviations was added
- 256. Please replace Figure 4 with a higher quality, higher resolution one.
Changed
- 354. Please replace Figure 5 with a higher quality, higher resolution one.
Fig. 5 was eliminated
- 372. Please replace Figure 6 with a higher quality, higher resolution one.
Changed
Reviewer 2 Report
The aim of this paper appears to be a literature survey to compile information concerning the antioxidant activities of different species, commonly referred to as “mistletoe”, plants exhibiting hemi parasitic and epiphytic characteristics, from around the world and their possible medical applications. However, there are taxonomic, species distribution, historical, folkloric medicinal elements that overshadow the antioxidant activities which are restricted only to the presence of reports in the literature and no comparative data or details are given in the text. The actual topic of the review is very interesting but the review itself has not been achieved. There is no in-depth analysis of the results presented in the bibliography, so the reader has not actually learnt the details but only knows what was studied by the researchers. In fact, of the 104 references given in the bibliography only 36 are related to antioxidant activity and are given in Table 1.
It should however be outlined how the literature search was carried out and which key words were used as an indication as to how comprehensive the study was.
The language is very descriptive and not concise scientific English. There are long passages with no reference to the bibliography. Try and simplify the language.
To be constructive, it is recommended that different sections could be revised to reorganize this review. For example, taxonomy, distribution, antioxidant activities, antioxidant compounds (composition), medical implementations presenting actual data from the research works cited. Outlining also the aims and future prospects for the use of mistletoes in medicine.
Otherwise change the emphasis of the review.
Some points are covered more specifically below:
Title The Antioxidant activity of mistletoes (Viscum album and related species)
It is better to use “other” species as related could mean taxonomic familiarity.
The title is not representative of the study as presented.
Introduction
The introduction is not very relevant to the topic and does not lay out the objectives of the study.
The introduction should start with information about mistletoes as implied in the title.
The first two paragraphs contain a lot of medical information which don’t appear to be very relevant to the immediate topic of the paper. It is suggested that they are abbreviated or omitted. The third paragraph could be expanded to include the objectives of the study or just start with the section on mistletoe plants.
Lines 32-34 “since they are decisive factors in the balance between the natural pathway to the final destination and a catastrophic anticipated end of the terrestrial transit.” Such sentences are very philosophical and should be stated more realistically.
Lines 44-48 the quote here can be omitted and by adding “These authors proposed that natural products……”.
Other text sections
Line 70 fronds do you mean branches?
Line 71 rich use enrich
Line 92 this section heading is not correct here. It also appears again on line 112.
Lines 93-93. Need references here.
Line 133 white not blank
Line 136 this paragraph is very general and talks about different laboratory methods but there are no references. Also not specifically related to mistletoe plants. More information can be obtained from the titles of the references in the bibliography rather than the text.
Line 258 C6C3 numbers as subscript
Line 341 survive not survey
Table 1. The mistletoe plants in relation to the related families. Relative? families.
The abbreviation Eu is not used in Figure 5
Only 36 references are given with antioxidant activity out of the 104 references given in the bibliography.
Figure 5. This figure does not show accurately the distribution of the different mistletoes only points are given with the letters.
There doesn’t appear to be a logical correspondence between the abbreviations and countries of distribution eg Q? Cr min or SA1 SA2
Is European mistletoe introduced or native to the countries outside of Europe?
Are there other mistletoes not included in this survey?
Figure 6. This figure if included should appear before the Conclusions. In its present form however, the coordination of activities is not evident. Delete or improve to be meaningful.
Check that species names/ scientific names are in italics throughout paper.
Conclusions
Authors should conclude whether the aims of the review have been achieved.
Lines 356 -8 “Table 1 contains the species reported in this review, including the common names, 356 the botanical family and the related references. A major direct information of their geo-357 graphical distribution can be deduced by Fig. 5.” This is not a conclusion.
References
Line 380 check the use of et al. in the list of references and be consistent.
Check all species names are in italics eg lines 391 401 and elsewhere.
Generally, correct according to Journal style.
Line 394 “hosts” not hots.
There are several grammatical errors.
The scientific style should be improved.
Author Response
The aim of this paper appears to be a literature survey to compile information concerning the antioxidant activities of different species, commonly referred to as “mistletoe”, plants exhibiting hemi parasitic and epiphytic characteristics, from around the world and their possible medical applications. However, there are taxonomic, species distribution, historical, folkloric medicinal elements that overshadow the antioxidant activities which are restricted only to the presence of reports in the literature and no comparative data or details are given in the text. The actual topic of the review is very interesting but the review itself has not been achieved. There is no in-depth analysis of the results presented in the bibliography, so the reader has not actually learnt the details but only knows what was studied by the researchers. In fact, of the 104 references given in the bibliography only 36 are related to antioxidant activity and are given in Table 1.
It should however be outlined how the literature search was carried out and which key words were used as an indication as to how comprehensive the study was.
I conducted an extensive literature survey mainly on Google, PubMed and Scopus following the Preferred Reporting Items for biological activity. The keywords were “Viscum, mistletoe, antioxidant, anti-ROS, and related genera”. The result is a personal data base of more than 400 references, wherein the most relevant were selected. The data base can be consulted. In my opinion, this information could be considered obvious, being the normal way to obtain a reliable review.
The language is very descriptive and not concise scientific English. There are long passages with no reference to the bibliography. Try and simplify the language.
Thanks for the suggestion. Several parts were changed in accordance and many references were added.
To be constructive, it is recommended that different sections could be revised to reorganize this review. For example, taxonomy, distribution, antioxidant activities, antioxidant compounds (composition), medical implementations presenting actual data from the research works cited. Outlining also the aims and future prospects for the use of mistletoes in medicine.
Otherwise change the emphasis of the review.
The proposed sequence is already present, with the exception that not only the medical implementations are considered for each species on the basis of the present information, not always complete and in consideration of the wide number of products and the tradition of medicinal plants. Some parts were reconsidered on the light of this suggestion. However, a complete revision of the architecture of the review, including its originality is considered not useful, because the review is dedicated to the complex world of natural products, nowadays and in the future, which is not only related to medicine. I would like to stress a point. This review is part of a special issue of the scientific journal Plants, whose content and typology where discussed before the initiative. However, the considerations of the reviewer are accepted, as possible and added to the conclusions and table 1 totally changed. Future prospects and aims have been added.
Some points are covered more specifically below:
Title The Antioxidant activity of mistletoes (Viscum album and related species)
It is better to use “other” species as related could mean taxonomic familiarity.
Changed in accordance
The title is not representative of the study as presented.
It is in accordance with items of the special issue. In any case, the paper was deeply revised to be more adherent to the title.
Reviewer N. 3
Introduction
The introduction is not very relevant to the topic and does not lay out the objectives of the study.
The introduction should start with information about mistletoes as implied in the title.
More information was added.
The first two paragraphs contain a lot of medical information which don’t appear to be very relevant to the immediate topic of the paper. It is suggested that they are abbreviated or omitted. The third paragraph could be expanded to include the objectives of the study or just start with the section on mistletoe plants.
Medical information is important, as evidenced by another reviewer, and considering the unique story of mistletoe, which is one of the topic of the research. The point is that information about a medicinal plant should be complete to understand the importance of these unique plants, from Botany, to Chemistry, Pharmacognosy and Medicine, in particular for a complex argument, like the antioxidant activity. It is noteworthy that the structure of this review is the same of the review already published in Plants about the anti-inflammatory activity of Viscum album.
Lines 32-34 “since they are decisive factors in the balance between the natural pathway to the final destination and a catastrophic anticipated end of the terrestrial transit.” Such sentences are very philosophical and should be stated more realistically.
Ok, the sentence was deleted. No problem, it can be used in another paper.
Lines 44-48 the quote here can be omitted and by adding “These authors proposed that natural products……”.
Changed in accordance
Other text sections
Line 70 fronds do you mean branches?
The term crown is surely better
Line 71 rich use enrich
Changed in reach
Line 92 this section heading is not correct here. It also appears again on line 112.
Changed in accordance
Lines 93-93. Need references here.
Line 133 white not blank
Changed in accordance
Line 136 this paragraph is very general and talks about different laboratory methods but there are no references. Also not specifically related to mistletoe plants. More information can be obtained from the titles of the references in the bibliography rather than the text.
Changed in accordance
Line 258 C6C3 numbers as subscript
It refers to the general formula of phenylpropanoids as derived from their biosynthetic origin from aromatic aminoacids. It could be changed in C6-C3, but the subscript has another meaning. In any case, the same consideration should inferred for C6C1, which could also changed in C6- As a reference on the matter, was added to clarify this aspect. Chang, Y.C.; et al. Free Radic Biol Med 2007
Line 341 survive not survey
Survey in the meaning of study, collection of data
Table 1. The mistletoe plants in relation to the related families. Relative? families.
Related in the meaning of taxonomically connected
The abbreviation Eu is not used in Figure 5
Fig. 5 was eliminated
Only 36 references are given with antioxidant activity out of the 104 references given in the bibliography.
Figure 5. This figure does not show accurately the distribution of the different mistletoes only points are given with the letters.
Fig. 5 was eliminated
There doesn’t appear to be a logical correspondence between the abbreviations and countries of distribution eg Q? Cr min or SA1 SA2
Is European mistletoe introduced or native to the countries outside of Europe?
Are there other mistletoes not included in this survey?
Fig. 5 was eliminated
Figure 6. This figure if included should appear before the Conclusions. In its present form however, the coordination of activities is not evident. Delete or improve to be meaningful.
The connection between the activities is already present in the text and in the cited references
Check that species names/ scientific names are in italics throughout paper.
Checked and changed
Conclusions
Authors should conclude whether the aims of the review have been achieved.
Added comments and information
Lines 356 -8 “Table 1 contains the species reported in this review, including the common names, 356 the botanical family and the related references. A major direct information of their geo-357 graphical distribution can be deduced by Fig. 5.” This is not a conclusion.
Fig. 5 was deleted.
References
Line 380 check the use of et al. in the list of references and be consistent.
Checked and changed
Check all species names are in italics eg lines 391 401 and elsewhere.
Changed, although the names of the plants were not in Italic in the original title of the reference
Generally, correct according to Journal style.
Changed
Line 394 “hosts” not hots.
Changed
Reviewer 3 Report
The author presents in detail, thoroughly and critically different species of mistletoe, classifying them according to their geographical distribution. The manuscript presents a complete overview of the use of Viscum species in the past and present, as well as gives future perspectives of their utilization. The author describes the botanical characteristics of Viscum species, illustrated with photographs, also their chemical composition, pharmacological effects and toxicity. However, the title of the paper suggests that the focus should be on the antioxidant effect of the reviewed plants. In the text, the author gives information about the antioxidant effect, the groups of the biologically active substances or specific compounds to which the antioxidant effect is/or it is supposed to be due. Nevertheless, this information is not reflected appropriately in the manuscript. My recommendation is the manuscript to be supplemented with figures or tables summarizing and illustrating the discussed antioxidant activity. In that way, the paper would be more informative for the readers. For example, the author could add some more columns to table 1, summarizing the different methods of antioxidant activity available for each mistletoe species, the substances responsible for these activities. Minor aspects: the Latin name Viscum in lines 201,205, 217..., should be in italic. The author presents in detail, thoroughly and critically different species of mistletoe, classifying them according to their geographical distribution. The manuscript presents a complete overview of the use of Viscum species in the past and present, as well as gives future perspectives of their utilization. The author describes the botanical characteristics of Viscum species, illustrated with photographs, also their chemical composition, pharmacological effects and toxicity. However, the title of the paper suggests that the focus should be on the antioxidant effect of the reviewed plants. In the text, the author gives information about the antioxidant effect, the groups of the biologically active substances or specific compounds to which the antioxidant effect is/or it is supposed to be due. Nevertheless, this information is not reflected appropriately in the manuscript. My recommendation is the manuscript to be supplemented with figures or tables summarizing and illustrating the discussed antioxidant activity. In that way, the paper would be more informative for the readers. For example, the author could add some more columns to table 1, summarizing the different methods of antioxidant activity available for each mistletoe species, the substances responsible for these activities. Minor aspects: the Latin name Viscum in lines 201,205, 217..., should be in italic.Author Response
The author presents in detail, thoroughly and critically different species of mistletoe, classifying them according to their geographical distribution. The manuscript presents a complete overview of the use of Viscum species in the past and present, as well as gives future perspectives of their utilization. The author describes the botanical characteristics of Viscum species, illustrated with photographs, also their chemical composition, pharmacological effects and toxicity. However, the title of the paper suggests that the focus should be on the antioxidant effect of the reviewed plants. In the text, the author gives information about the antioxidant effect, the groups of the biologically active substances or specific compounds to which the antioxidant effect is/or it is supposed to be due. Nevertheless, this information is not reflected appropriately in the manuscript. My recommendation is the manuscript to be supplemented with figures or tables summarizing and illustrating the discussed antioxidant activity. In that way, the paper would be more informative for the readers. For example, the author could add some more columns to table 1, summarizing the different methods of antioxidant activity available for each mistletoe species, the substances responsible for these activities. Minor aspects: the Latin name Viscum in lines 201,205, 217..., should be in italic.
The text was deeply revised considering the recommendations, including in particular a table 1 completely revised and this was preferred instead add information to the table.
Round 2
Reviewer 1 Report
Many thanks to the author for eliminating all the comments.
Author Response
Many thanks for your comments to improve the MN
Reviewer 2 Report
The manuscript has generally been revised according to suggestions.
Please consider the following points:
1.Author: “an extensive literature survey mainly on Google, PubMed and Scopus following the Preferred Reporting Items for biological activity. The keywords were “Viscum, mistletoe, antioxidant, anti-ROS, and related genera”.
It would be beneficial for the reader to include this information in the text.
2.Author: It is noteworthy that the structure of this review is the same of the review already published in Plants about the anti-inflammatory activity of Viscum album.
This review is by the same author and may contain overlapping information.
3.Author: The term crown is surely better. Yes, use it.
4.Line 341 survive not survey. Author: Survey in the meaning of study, collection of data.
The sentence should be checked for meaning. In its present form survey doesn’t fit.
5.Figure 6. This figure if included should appear before the Conclusions. In its present form however, the coordination of activities is not evident. Delete or improve to be meaningful.
Author: The connection between the activities is already present in the text and in the cited references
Then the figure is not necessary. There are no arrows indicating connectivity.
6.Text
Line 329 triterpenic
Line 428-9 it wasn’t clear in the revised text if Figure 5 was deleted as it is referred to in these lines and also lines 447-8.
Lines 446-7 check if botanical Family is given in Table 1
English language of the text should be edited to help comprehension.
Author Response
The manuscript has generally been revised according to suggestions.
Please consider the following points:
1.Author: “an extensive literature survey mainly on Google, PubMed and Scopus following the Preferred Reporting Items for biological activity. The keywords were “Viscum, mistletoe, antioxidant, anti-ROS, and related genera”.
It would be beneficial for the reader to include this information in the text.
The information was included.
2.Author: It is noteworthy that the structure of this review is the same of the review already published in Plants about the anti-inflammatory activity of Viscum album.
This review is by the same author and may contain overlapping information.
The comment was considered and some parts changed in accordance.
3.Author: The term crown is surely better. Yes, use it.
In accordance.
4.Line 341 survive not survey. Author: Survey in the meaning of study, collection of data.
The sentence should be checked for meaning. In its present form survey doesn’t fit.
Changed in accordance. Thanks.
5.Figure 6. This figure if included should appear before the Conclusions. In its present form however, the coordination of activities is not evident. Delete or improve to be meaningful.
Author: The connection between the activities is already present in the text and in the cited references
Then the figure is not necessary. There are no arrows indicating connectivity.
The figure was deleted.
6.Text
Line 329 triterpenic
Changed in accordance
Line 428-9 it wasn’t clear in the revised text if Figure 5 was deleted as it is referred to in these lines and also lines 447-8.
Corrected. Thanks
Lines 446-7 check if botanical Family is given in Table 1
This information was deleted from the table, being already reported in the text for each species.